# Effects of amotosalen treatment on human platelet lysate bioactivity: A proof-of-concept study

Christian Christensen[1,2,3], Sandra Mjoll Jonsdottir-Buch[1,2,3], Olafur Eysteinn Sigurjonsson [1,2,3,4]*

1 The Blood Bank, The National University Hospital of Iceland, Reykjavik, Iceland, 2 Faculty of Medicine, Biomedical Center, University of Iceland, Reykjavik, Iceland, 3 Platome Biotechnology, Hafnarfjörður, Iceland, 4 School of Science and Engineering, University of Reykjavik, Reykjavik, Iceland

* oes@landspitali.is

## Abstract

### Background

Clinical application of mesenchymal stromal cells (MSCs) usually requires an *in vitro* expansion step to reach clinically relevant numbers. *In vitro* cell expansion necessitates supplementation of basal mammalian cell culture medium with growth factors. To avoid using supplements containing animal substances, human platelet lysates (hPL) produced from expired and pathogen inactivated platelet concentrates can be used in place of fetal bovine serum. However, globally, most transfusion units are currently not pathogen inactivated. As blood banks are the sole source of platelet concentrates for hPL production, it is important to ensure product safety and standardized production methods. In this proof-of-concept study we assessed the feasibility of producing hPL from expired platelet concentrates with pathogen inactivation applied after platelet lysis by evaluating the retention of growth factors, cytokines, and the ability to support MSC proliferation and tri-lineage differentiation.

### Methodology/Principal findings

Bone marrow-derived MSCs (BM-MSCs) were expanded and differentiated using hPL derived from pathogen inactivated platelet lysates (hPL-PIPL), with pathogen inactivation by amotosalen/ultraviolet A treatment applied after lysis of expired platelets. Results were compared to those using hPL produced from conventional expired pathogen inactivated platelet concentrates (hPL-PIPC), with pathogen inactivation applied after blood donation. hPL-PIPL treatment had lower concentrations of soluble growth factors and cytokines than hPL-PIPC treatment. When used as supplementation in cell culture, BM-MSCs proliferated at a reduced rate, but more consistently, in hPL-PIPL than in hPL-PIPC. The ability to support tri-lineage differentiation was comparable between lysates.

### Conclusion/Significance

These results suggest that functional hPL can be produced from expired and untreated platelet lysates by applying pathogen inactivation after platelet lysis. When carried out post-

**Data Availability Statement:** All relevant data are within the paper.

**Funding:** The funder provided support in the form of salaries for authors [SMJ], but did not have any

additional role in the study design, data collection and analysis, decision to publish, or preparation of the manuscript. The specific roles of these authors are articulated in the 'author contributions' section.

**Competing interests:** OES and SJM are are shareholders in Platome biotechnology (PB). SJM is the CEO of PB and recived salaries from PB during the study. CC holds no shares and was not paid by BP during the study. This does not alter our adherence to PLOS ONE policies on sharing data and materials.

expiration, pathogen inactivation may provide a valuable solution for further standardizing global hPL production methods, increasing the pool of starting material, and meeting future demand for animal-free supplements in human cell culturing.

## Introduction

Pathogen inactivation systems are proactive alternatives to conventional bacterial screening and prevention methods for blood transfusion products [1]. Several systems are currently available on the market such as the INTERCEPT™ Blood System, Mirasol™ and THERAFLEX™ [2]. The INTERCEPT™ Blood System for pathogen inactivation of platelets gained the CE mark in 2002 [3] and is routinely used in several European countries [4]. The system effectively reduces the accumulation of pathogens by blocking their ability to replicate [3]. The active compound used in the INTERCEPT System, amotosalen S-59, passes through cell membranes, bacterial walls, and viral envelopes and intercalates between helical regions of DNA and RNA [5]. Covalent crosslinks are formed between amotosalen S-59 and pyrimidine bases upon exposure to ultraviolet A (UVA) illumination, leaving most pathogenic organisms unable to replicate [3]. While successful inactivation is observed in general, non-enveloped viruses such as hepatitis A and hepatitis E show resistance to the treatment [3].

Implementing a pathogen inactivation system improves the safety of transfusion units and allows platelets to be stored for an extended period of up to seven days [3]. Blood banks must stock surplus transfusion units; with approximately 2 million reported annual platelet transfusions in the United States alone [6], this inevitably results in large quantities of expired platelets [7]. Although expired platelet concentrates are unfit for transfusion medicine due to safety concerns, their abundance of growth factors makes them suitable for use in cell cultures [8].

Human mesenchymal stromal cells (MSCs) are multipotent cells capable of undergoing *in vivo* differentiation into end-stage cell types of connective tissues [9]. They were initially described, in the 1960s, as plastic-adherent and fibroblast-like cells with clonogenic potential *in vitro* [10]. Today, they are recognized as a part of stem cell niches in the bone marrow [11] and are known to play a role in immunomodulation [12], as demonstrated by the prominent contribution of the cells in reversing graft-versus-host-disease [13]. Scientific interest in MSCs has developed in recent years, making them one of the most-studied human cell types related to cell-based therapy [14].

An *in vitro* expansion step is usually necessary to attain clinically relevant numbers of MSCs [15]. Successful expansion of MSCs requires that basal cell culture medium be supplemented with a source of mitogens, such as growth factors. To date, this has mostly been performed using fetal bovine serum (FBS), which contains low amounts of immunoglobins and complement factors [16]. However, concerns regarding the content of animal components and large lot-to-lot variability has initiated a quest for replacements [16–18]. Platelet derivatives such as human platelet lysates (hPL) are currently considered promising replacements for FBS [8,19–22] due to their non-xenogeneic nature and abundance of growth factors [23]. hPL has successfully been produced from fresh platelets [24], expired platelets [8], and, most recently, from expired and pathogen inactivated platelets [22]; all have been found to either be comparable to or to outperform FBS.

hPL is commonly manufactured by exposing platelet concentrates to repetitive cycles of freezing and thawing, resulting in degranulation [21], but it can also be produced by $CaCl_2$ activation [25], sonication [26], or by using a solvent/detergent strategy [27]. Degranulation

causes the α-granules to release growth factors and cytokines into solution [28]. To obtain a sufficient volume of hPL, allogeneic hPL are typically pooled using up to 120 donors [29]; however, recent research has demonstrated that pooling can be done effectively using up to 250 donors [30]. The end product is refined by centrifugation and sterile filtration to remove platelet fragments [29].

Recently, emphasis has been placed on improving standardization in global good manufacturing practice (GMP)-grade hPL production, as current production methods vary between manufacturers [31]. Although pathogen inactivation techniques have recently been introduced into the preparation of transfusion products, they are predominantly applied in Europe, while blood banks in the United States and Asia primarily rely on bacterial screening systems such as BacT/ALERT to ensure product safety [32]. As a result, most platelets used for hPL production are currently not pathogen inactivated.

In this proof-of-concept study, hPL from pathogen inactivated platelet lysates (hPL-PIPL), where pathogen inactivation was applied after lysis of expired platelets, was compared to hPL produced from expired conventionally pathogen inactivated platelet concentrates (hPL-PIPC), where pathogen inactivation was applied after production of platelet concentrates. We evaluated and compared the total protein content and concentrations of selected soluble growth factors and cytokines between hPL-PIPL and hPL-PIPC. Furthermore, we compared long-term proliferation and tri-lineage differentiation of bone marrow-derived MSCs using hPL-PIPL and hPL-PIPC as cell culture media supplements in order to determine the feasibility of producing an effective cell culture supplement from hPL when pathogen inactivation is applied after platelet lysis.

## Results

### Growth factor and cytokine concentrations are lower in hPL-PIPL than in hPL-PIPC

Quantification of 37 soluble growth factors and cytokines was performed using Luminex xMAP technology following platelet lysate production. Each of the measured growth factors and cytokines was present at lower concentrations in hPL-PIPL than in hPL-PIPC (Table 1). The mean decreases for all evaluated growth factors and cytokines in hPL-PIPL (as compared to hPL-PIPC) for two produced batches were $29 \pm 15\%$ ($p < 0.001$) and $36 \pm 19\%$ ($p < 0.001$). The mean differences between hPL-PIPL and hPL-PIPC in key growth factors for the two batches ranged from 4.2 to 36%: platelet-derived growth factor-AB/BB (PDGF-AB/BB) ($4.2 \pm 1.8\%$); platelet-derived growth factor-AA (PDGF-AA) ($12 \pm 17\%$), epidermal growth factor (EGF) ($23.9 \pm 7.8\%$); vascular endothelial growth factor (VEGF) ($26.6 \pm 0.62\%$); and basic fibroblast growth factor (FGF-2) ($36.3 \pm 1.8\%$).

### Proliferation of MSCs in hPL-PIPL is comparable to, but slower than, that in hPL-PIPC

Proliferation of MSCs originating from the bone marrow of two individual donors were assessed during nine cell passages (Fig 1). Cumulative population doublings (CPDs) were assessed from passage 4 through passage 9. MSCs grown in cell culture media supplemented with hPL-PIPL (hPL-PIPL-MSCs) proliferated significantly slower than hPL-PIPC-MSCs ($p < 0.001$) from passage 5 through passage 9. At the end of passage 9, MSCs from Donor 6 had reached $19.42 \pm 0.10$ CPDs and $21.25 \pm 0.03$ CPDs with hPL-PIPL and hPL-PIPC supplementation, respectively, and MSCs from Donor 13 had reached $10.50 \pm 0.40$ CPDs and $12.80 \pm 0.20$ CPDs.

**Table 1. Growth factor and cytokine concentrations in two batches of undiluted hPL.**

| Growth Factor / Cytokine | Batch 23 | | | Batch 24 | | | Batch 23 vs. Batch 24 |
|---|---|---|---|---|---|---|---|
| | hPL-PIPC | hPL-PIPL | Difference[a] | hPL-PIPC | hPL-PIPL | Difference | Mean ± SD[b] |
| | pg/mL | pg/mL | % | pg/mL | pg/mL | % | % |
| EGF | 1992 | 1361 | 32 | 346 | 291 | 16 | 22.9 ± 7.8 |
| Eotaxin | 96 | 53 | 45 | 129 | 88 | 32 | 38.2 ± 6.7 |
| FGF-2 | 517 | 320 | 38 | 478 | 313 | 34 | 36.3 ± 1.8 |
| Fractalkine | 234 | 196 | 16 | 164 | 105 | 36 | 26 ± 10 |
| G-CSF | 49 | 27 | 44 | 57 | 27 | 53 | 48.5 ± 4.1 |
| GM-CSF | 27 | 20 | 26 | 27 | 14 | 48 | 37 ± 11 |
| IFNα2 | 76 | 49 | 35 | 84 | 44 | 47 | 41.2 ± 6.2 |
| IFNγ | 14 | 12 | 19 | 14 | 8.5 | 40 | 29 ± 11 |
| IL-1α | 59 | 50 | 16 | 29 | 19 | 35 | 25.4 ± 9.6 |
| IL-1β | 5.5 | 3.4 | 38 | 6.4 | 2.6 | 59 | 48 ± 11 |
| IL-1RA | 503 | 452 | 10 | 1030 | 884 | 14 | 12.2 ± 2.0 |
| IL-2 | 7.8 | 6.1 | 23 | 4.0 | 1.8 | 55 | 39 ± 16 |
| IL-3 | 6.3 | 4.3 | 32 | 6.2 | 3.1 | 49 | 40.6 ± 8.4 |
| IL-4 | 31 | 17 | 47 | 48 | 27 | 44 | 45.5 ± 1.1 |
| IL-5 | 34 | 28 | 16 | 82 | 69 | 16 | 15.7 ± 0.0 |
| IL-6 | 13 | 7.9 | 40 | 12 | 3.3 | 72 | 56 ± 16 |
| IL-7 | 30 | 20 | 32 | 36 | 22 | 40 | 36.4 ± 3.9 |
| IL-8 | 50 | 38 | 24 | 76 | 64 | 16 | 19.9 ± 3.8 |
| IL-9 | 10 | 7.2 | 29 | 6.5 | 3.9 | 39 | 34.3 ± 5.0 |
| IL-10 | 20 | 14 | 30 | 7 | 2.4 | 65 | 48 ± 18 |
| IL-12p40 | 61 | 51 | 16 | 62 | 32 | 49 | 33 ± 16 |
| IL-12p70 | 11 | 6.2 | 44 | 12 | 6.2 | 47 | 45.5 ± 1.5 |
| IL-13 | 183 | 176 | 3.7 | 433 | 372 | 14 | 8.8 ± 5.2 |
| IL-15 | 12 | 6.6 | 45 | 8.6 | 3.6 | 59 | 51.6 ± 7.1 |
| IL-17A | 14 | 14 | 4.1 | 10 | 5.5 | 43 | 23 ± 19 |
| IP-10 | 120 | 58 | 52 | 101 | 51 | 49 | 50.7 ± 1.4 |
| MCP-1 | 157 | 111 | 30 | 185 | 157 | 15 | 22.4 ± 7.4 |
| MCP-3 | 301 | 276 | 8.3 | 520 | 491 | 5.7 | 7.0 ± 1.3 |
| MDC | 705 | 192 | 73 | 754 | 230 | 69 | 71.1 ± 1.7 |
| MIP-1α | 24 | 22 | 11 | 33 | 30 | 8.8 | 9.8 ± 1.0 |
| MIP-1β | 123 | 82 | 34 | 165 | 127 | 23 | 28.4 ± 5.4 |
| PDGF-AA | 15850 | 11232 | 29 | 9105 | 9548 | -4.9 | 12 ± 17 |
| PDGF-AB/BB | 27802 | 27146 | 2.4 | 25157 | 23661 | 5.9 | 4.2 ± 1.8 |
| TGF-α | 3.0 | 1.8 | 40 | 4.1 | 1.9 | 53 | 46.5 ± 6.5 |
| TNFα | 24 | 16 | 35 | 26 | 15 | 42 | 38.5 ± 3.5 |
| TNFβ | 266 | 236 | 11 | 570 | 490 | 14 | 12.7 ± 1.4 |
| VEGF | 538 | 392 | 27 | 441 | 327 | 26 | 26.6 ± 0.6 |
| Mean ± SD (%) | | | 29 ± 15 | | | 36 ± 19 | 32 ± 16 |
| p-value[c] | | | < 0.001 | | | < 0.001 | < 0.001 |

[a] Difference (%) between hPL-PIPC and hPL-PIPL within individual batches (Batch 23 and Batch 24). Note that the concentrations for all growth factors and cytokines were lower in hPL-PIPL.

[b] Mean ± SD (%) represents the mean difference between hPL-PIPC and hPL-PIPL for both batches (n = 4).

[c] p-values are reported on the overall differences between treatments (hPL-PIPC and hPL-PIPL) in individual batches (23 and 24) and for both batches combined. p-values were determined using a paired ratio t-test.

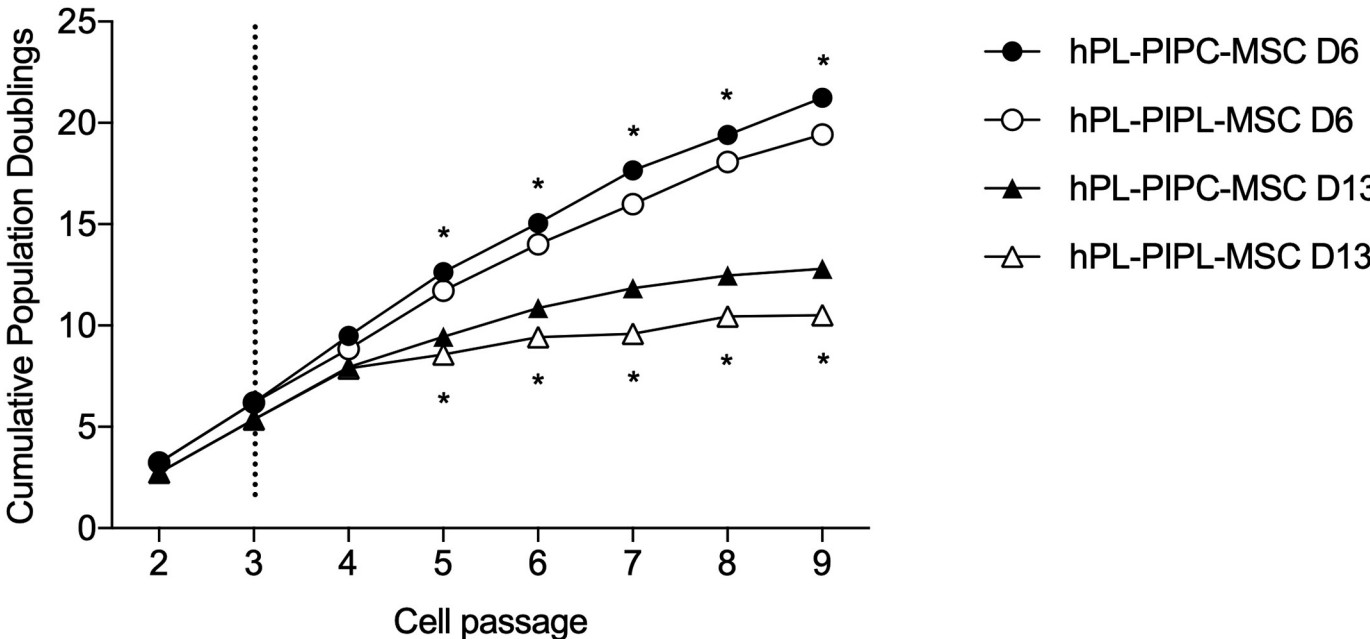

**Fig 1. Long-term proliferation of MSCs, expressed as cumulative population doublings.** Proliferation of MSCs from two donors (D6 and D13) was evaluated in cell culture media supplemented with hPL-PIPC or hPL-PIPL after passage 3 (indicated by the vertical dotted line). Points represent mean ± SEM at the end of each passage (n = 6 cell cultures per passage, assessed in two independent experiments). Asterisks (* p < 0.05) indicate statistical significance between hPL-PIPC and hPL-PIPL for an individual donor, evaluated via a two-way ANOVA with Tukey's post hoc test.

### Tri-lineage differentiation is not affected by the timing of pathogen inactivation

hPL-PIPL-MSCs were successfully differentiated into osteogenic, adipogenic and chondrogenic lineages. Osteogenic differentiation was evaluated during 28 days of stimulation in osteogenic media. Alkaline phosphatase (ALP) activities in both hPL-PIPL-MSCs and hPL-PIPC-MSCs were significantly elevated after seven days as compared to unstimulated control cultures (p < 0.05) (Fig 2). A peak in ALP activity was observed at day 14 for both hPL-PIPL-MSCs (3.6 ± 0.1 nmol (p-nitrophenol)/min) and hPL-PIPC-MSCs (3.8 ± 0.1 nmol (p-nitrophenol)/min), followed by fairly consistent ALP activity between days 14 and 28. Comparable mineralization between hPL-PIPC and hPL-PIPL MSCs was demonstrated by Alizarin Red S staining after 28 days of differentiation (Fig 3A and 3B).

Adipogenic differentiation was evaluated during 14 days of stimulation in adipogenic media. Accumulation of lipid droplets in the cell periphery was confirmed by positive Oil Red O staining after 7 days. After 14 days of differentiation, lipid droplets were distributed throughout the cells (Fig 3C and 3D).

Chondrogenic differentiation was evaluated during 35 days of stimulation in chondrogenic media. After 28 days of differentiation, the concentration of glycosaminoglycans (GAGs) was significantly higher (p < 0.05) in cell pellets supplemented with both hPL-PIPC and hPL-PIPL than in unstimulated control cell pellets (Fig 4). GAG concentration remained significantly higher than the control during differentiation of hPL-PIPC-MSCs at day 35 (p < 0.05), while GAG concentration decreased again in hPL-PIPL-MSCs by day 35 such that there was no longer a significant difference from the control. No statistically significant differences were observed between hPL-PIPC-MSCs and hPL-PIPL-MSCs. Lacunae formation and accumulation of collagen fibers were demonstrated after 35 days of differentiation using Masson's trichrome staining (Fig 3E and 3F).

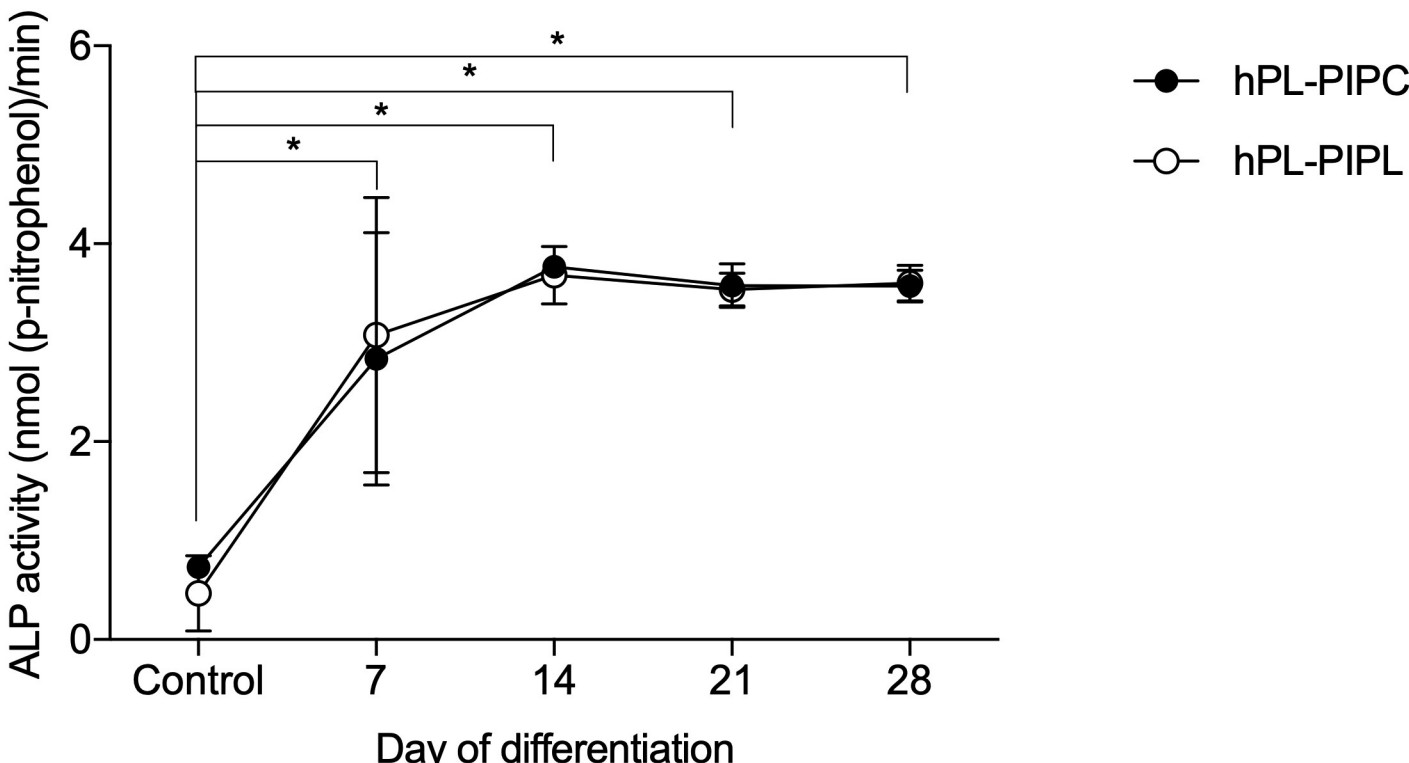

**Fig 2. Alkaline phosphatase activity during osteogenic differentiation.** MSCs were differentiated in osteogenic media supplemented with hPL-PIPL or hPL-PIPC, with MSCs grown in expansion media included as a control. Points represent means ± SEM (n = 6 cell cultures per timepoint, assessed in two independent experiments). Asterisks ($^*$ $p < 0.05$) indicate statistical significance versus the control, evaluated via a two-way ANOVA with Tukey's post hoc test.

## Discussion

In this proof-of-concept study, the feasibility of producing cell culture supplements from hPL derived from pathogen inactivated platelet lysates (hPL-PIPL) was evaluated via a comparison with a supplement created from conventional hPL derived from pathogen inactivated platelet concentrates (hPL-PIPC). We evaluated the composition of the lysates as well as their applicability for use as supplements to support BM-MSC proliferation and tri-lineage differentiation. As it has previously been shown that hPL performance as a cell culture supplement is comparable to or better than that of FBS [8, 22, 24], FBS was not included in this proof-of-concept work.

Concentrations of all 37 selected soluble growth factors and cytokines were significantly reduced in hPL-PIPL compared to hPL-PIPC, with an average difference of 32 ± 16% (p < 0.001). Of the key growth factors, we found that the differences between hPL-PIPC and hPL-PIPL in PDGF-AB/BB and PDGF-AA were relatively small, at 4.2 ± 1.8% and 12 ± 17%, respectively. The effects on EGF, VEGF, and FGF-2 were slightly higher, with observed reductions in hPL-PIPL of 23.9 ± 7.8%, 26.57 ± 0.62, and 36.3 ± 1.8%, respectively. As several growth factors within the α-granules of the platelets are important for MSC proliferation and differentiation [33,34], the composition following production and storage is an important marker for platelet lysate quality. The effects of pathogen inactivation on growth factor stability during storage have previously been studied. It was demonstrated that UVC treatment of platelet concentrates had no effect on concentrations of EGF, FGF-2, PDGF-AB, VEGF, or insulin-like growth factor (IGF) [35]. In a study specifically conducted on the INTERCEPT™ Blood System

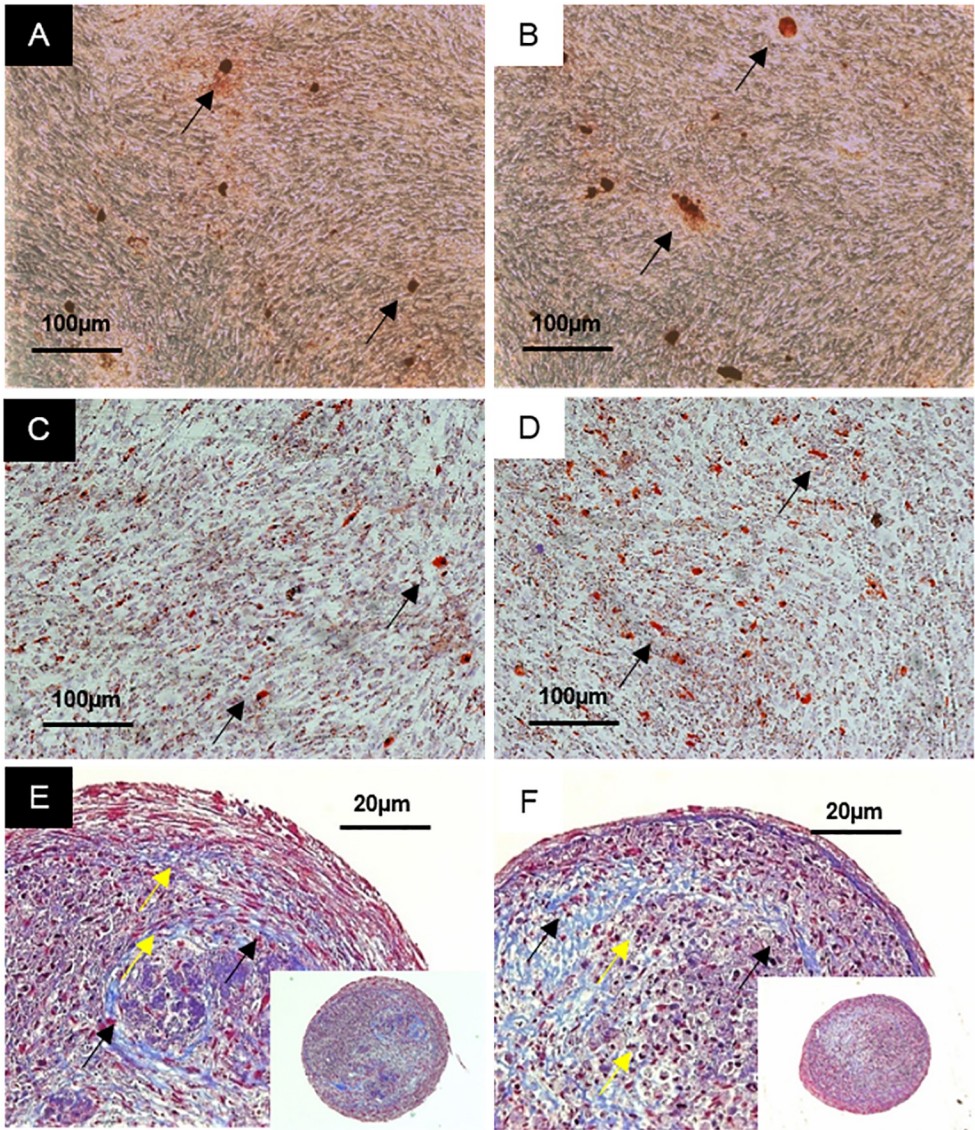

**Fig 3. Tri-lineage differentiation of MSCs.** MSCs differentiated using hPL-PIPC are shown in A, C, and E, while MSCs differentiated using hPL-PIPL are shown in B, D, and F. A and B show Alizarin Red S staining, used to demonstrate mineralization (black arrows) after 28 days of stimulation in osteogenic medium. C and D show Oil Red O staining, used to demonstrate accumulation of lipid droplets (black arrows) after 14 days of stimulation in adipogenic medium. E and F show Masson's trichrome staining, used to demonstrate collagen fibers (black arrows) and lacunae formation (yellow arrows) after 35 days of chondrogenic stimulation.

for pathogen inactivation, it was found that amotosalen plus UVA treatment mainly targets proteins of intracellular platelet activation pathways [36]. In addition, UV illumination of platelets combined with riboflavin or amotosalen seems to trigger activation of p38 mitogen-activated protein kinases (p38MAPK), leading to platelet degranulation [37]. However, to our knowledge this is the first study evaluating the effect of pathogen inactivation on lysed platelets compared to intact platelet concentrates. It can be speculated that, as a result of platelet cargo being released into solution of hPL-PIPL at the time of pathogen inactivation, the difference in composition between hPL-PIPC and hPL-PIPL is caused by UVA photodegradation, since

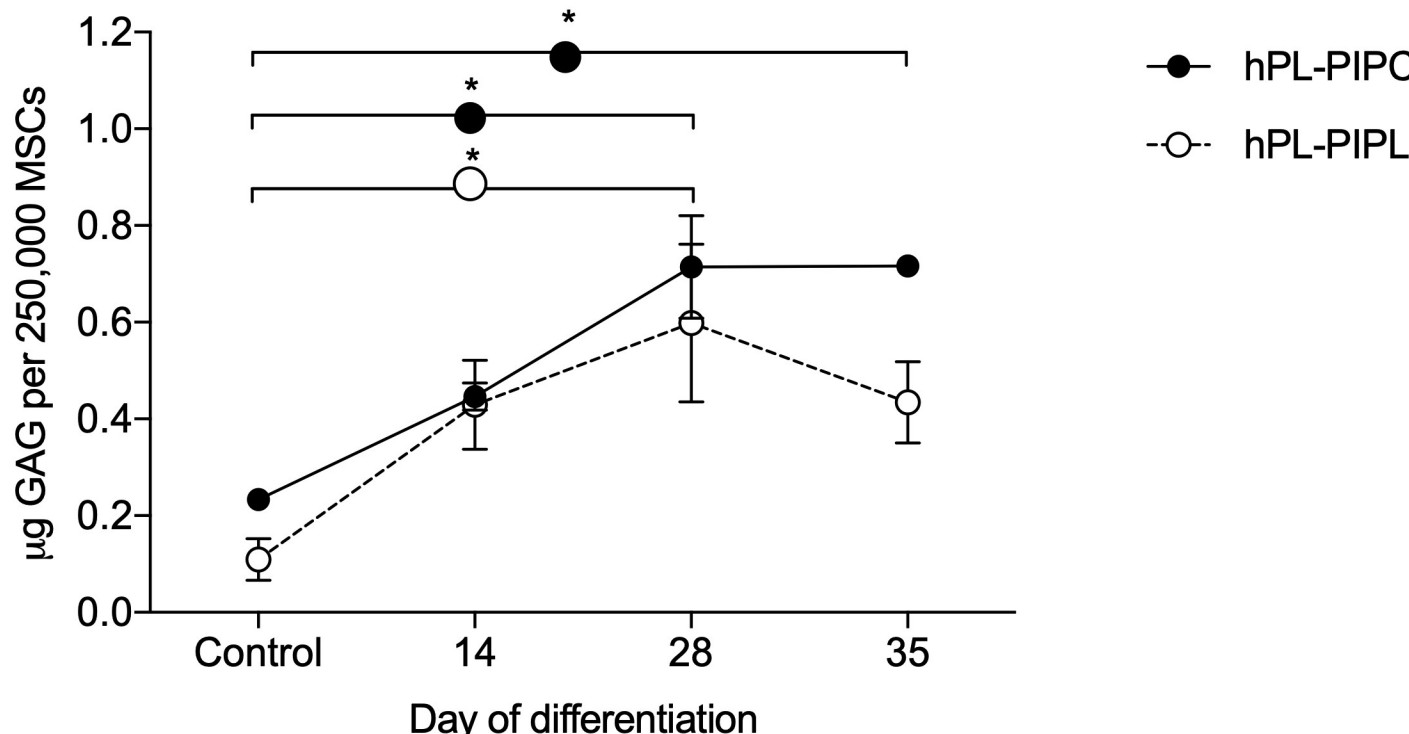

**Fig 4. Concentration of glycosaminoglycans.** GAG concentration was measured in MSC pellets stimulated with chondrogenic media supplemented with either hPL-PIPC or hPL-PIPL. Pellets grown in expansion media were included as a control. Points represent mean ± SEM (n = 2 pellets per timepoint, assessed in two individual experiments). Asterisks (* $p < 0.05$) indicate statistical significance versus the control, evaluated via a two-way ANOVA with Tukey's post hoc test.

light exposure can lead to irreversible changes in the primary, secondary, and tertiary structure of proteins [38]. It should be noted that the individual differences in soluble growth factors and cytokines between the two batches produced in this study limits the ability to generalize the results of changes to growth factors and cytokines beyond this proof-of-concept study.

As it has been suggested that hPL should contain high concentrations of PDGF-AB, VEGF, EGF, FGF-2, and transforming growth factor beta 1 (TGF-β1), and low concentrations of IGF-1, we examined the long-term proliferation of MSCs from two donors in terms of cumulative population doublings (CPDs). Total CPDs were higher for both of the donor hPL-PIPC-MSCs than for the hPL-PIPL-MSCs, indicating higher mitogenic effects in conventional hPL-PIPC. It is evident that EGF and VEGF activate the Raf-MEK-ERK pathway by binding to transmembrane receptor proteins at the plasma membrane [39]. Less activation of the Raf-MEK-ERK pathway may contribute to less cell proliferation and failure to prevent apoptosis. However, it is important to mention that high CPDs should not be viewed as a success criterion in isolation; successful expansion of MSCs must also include retention of genomic stability and avoidance of tumorigenicity [40]. These parameters were not within the scope of this study, but to fully elucidate the ability of hPL-PIPL to support long-term proliferation and clinical potential of MSCs *in vitro*, genomic stability and tumorigenicity is important. Future work should consider these factors.

Tri-lineage differentiation potential was examined *in vitro* by stimulating MSCs in osteogenic, adipogenic, and chondrogenic media. After 14 days of differentiation, both hPL-PIPC-MSCs and hPL-PIPL-MSCs differentiating into osteoblasts demonstrated significantly higher ALP activity compared to unstimulated control cultures. The presence of peak ALP levels around day

14 is a marker for osteogenic differentiation [41] and, in our study, indicated osteogenic differentiation potential. This was supported by mineralization and bone-like nodule formation after 28 days of differentiation in both treatments.

Adipogenic differentiation potential was evaluated over the course of 14 days. As mature adipocytes predominantly consist of lipid droplets [42], we used positive Oil Red O staining to visualize the transition from MSCs into adipocytes. Morphological alterations from spindle-shaped MSCs toward round adipocytes were observed after 7 days in both hPL types. Similarly, lipid droplets had formed at the periphery of the cells by day 7, and after 14 days the lipid droplets took up most of the intracellular space.

Finally, we evaluated chondrogenic differentiation potential over a period of 35 days. Cell pellets from both hPL-PIPC-MSCs and hPL-PIPL-MSCs had accumulated significantly more glycosaminoglycans (GAGs) after 28 days of differentiation than unstimulated control cultures. This was consistent at day 35 for hPL-PIPC-MSCs, whereas the GAG concentration decreased between days 28 and 35 for hPL-PIPL-MSCs. To evaluate changes within pellet structures, the pellets were sectioned and stained with Masson's trichrome. During differentiation, we observed lacunae formation and collagen fiber formation, which confirmed successful chondrogenic differentiation.

Based on the preliminary results obtained from this study, it appears that the application of pathogen inactivation techniques after platelet expiry and lysate production is possible and may prove to be a valuable tool in the pursuit of optimal safety and standardization in therapeutic-grade human platelet lysate production. Despite decreased growth factors and cytokines in hPL-PIPL compared to hPL-PIPC, levels were shown to be sufficient in hPL-PIPL to support MSC proliferation and differentiation into osteoblasts, adipocytes, and chondrocytes.

The results of this work imply that further research into the suitability of applying pathogen inactivation to lysates from expired platelets is warranted. Given that this was a proof-of-concept study, several limitations should be addressed in further research. Firstly, a larger number of hPL batches should be evaluated in order to better evaluate the changes in growth factors and cytokines based on the hPL production method, as there were large differences between the two batches used in this study. A comparison to an FBS control would also be warranted, in order to compare the levels of growth factors and cytokines in hPL-PIPL supplements to those found in FBS supplements. In addition, it would be valuable to examine the effect of hPL production method on genomic stability and tumerogenicity.

## Conclusion

In this proof-of-concept study, we demonstrated that functional hPL cell culture supplements can be produced by performing pathogen inactivation after platelet lysis of expired and previously untreated platelet concentrates (hPL-PIPL). hPL-PIPL supported long-term cell proliferation and tri-lineage differentiation of BM-MSCs. While hPL-PIPL performed comparably to hPL-PIPC in terms of tri-lineage differentiation, lower cumulative population doublings (CPDs) were observed for hPL-PIPL. hPL-PIPL was also found to contain lower concentrations of key growth factors, suggesting that the timing of pathogen inactivation may affect the mitogenic potential of hPL rather than differentiation potential. Further work is necessary to evaluate the practical implications of the differences between hPL-PIPL, hPL-PIPC, and FBS on the effectiveness of hPL-PIPL as a cell culture supplement.

## Materials and methods

### Preparation of platelet lysates

Study was approved by National Bioethics committee number VSN19-189. Four platelet concentrates (PCs) were prepared from a total of 32 buffy coats according to standard procedure

at the Blood Bank, Landspitali (The National University Hospital of Iceland), Reykjavík, Iceland, as specified in Table 2. Two separate batches of PCs (batch nos. 23 and 24) were each prepared by pooling two buffy coat-derived PCs together. Each pooled batch represented 16 whole blood donations obtained from healthy donors of the Blood Bank. Each batch was further split into two units and exposed to pathogen inactivation at different timepoints. The first unit, hPL-PIPC, was exposed to pathogen inactivation less than 24 hours post donation using the INTERCEPT™ Blood System (Cerus Corporation, Concord, CA, USA), which involves exposure to amotosalen S-59 and UVA illumination, according to manufacturer's protocol. Following pathogen inactivation, the PCs were placed into a platelet agitator at 22 ± 2˚C for seven days until expiration, and then stored at -80˚C for three weeks prior to platelet lysate production. The second unit, hPL-PIPL, was placed directly into the platelet agitator at 22 ± 2˚C without being pathogen inactivated, stored for seven days until expiration, and then transferred to -80˚C storage for three weeks. Pathogen inactivation for this second unit was performed after platelet lysis.

The expired PCs were collected and subjected to platelet lysis by three cycles of thawing at 37˚C and freezing at -80˚C to initiate degranulation. After the third cycle, the platelet lysates were aliquoted in 50-mL centrifugation tubes (Corning Science, Reynosa Tamaulipas, Mexico). Platelet fragments were removed by centrifugation at 4975 × g for 20 minutes using a Heraeus Multifuge X3 (Thermo Scientific, Waltham, Massachusetts, USA). This centrifugation step was repeated after the supernatants (platelet lysates) from each tube were transferred to new 50-mL centrifugation tubes. Prepared lysates from each unit were pooled. The hPL-PIPC units were distributed into 45 mL aliquots and stored at -20˚C in a freezer (Gram BioLine, Vojens, Denmark). In addition, 1 mL and 5 mL aliquots were prepared for composition analysis and adipogenic differentiation, respectively, and stored at -20˚C prior to analysis. The hPL-PIPL units were injected into sterile bags (Cerus Corporation, Concord, CA, USA) and exposed to pathogen inactivation with the INTERCEPT™ Blood System, according to the manufacturer's protocol, before being aliquoted and stored at -20˚C as above.

These undiluted platelet lysates were used for experimentation within 18 months of storage.

## Growth factor and cytokine quantification

The undiluted platelet lysates were analyzed using Luminex xMAP Technology (EMD Millipore Corporation, Billerica, MA, USA) to quantify 37 soluble growth factors and cytokines. The Human Cytokine/Chemokine Magnetic Bead Panel (HCYTOMAG-60K, Millipore) was used; it applies microspheres and fluorescent signaling to quantify EGF, Eotaxin, FGF-2, Fractalkine, G-CSF, GM-CSF, IFNα2, IFNγ, IL-1α, IL-1β, IL-2, IL-3, IL-4, IL-5, IL-6, IL-7, IL-8, IL-9, IL-10, IL-12P40, IL-12P70, IL-13, IL-15, IL-17A, IL-1RA, IP-10, MCP-1, MCP-3, MDC,

**Table 2. Platelet concentrate characteristics.**

| PC No. | Number of buffy coats[a] | | | Mean platelet count (× $10^9$) | Batch no. |
|---|---|---|---|---|---|
| | Total | O+ | O- | | |
| 1 | 8 | 7 | 1 | 278 ± 88.6 | 23 |
| 2 | 8 | 4 | 4 | 211 ± 50.5 | 23 |
| 3 | 8 | 8 | 0 | 197 ± 40.2 | 24 |
| 4 | 8 | 5 | 3 | 195 ± 25.4 | 24 |
| Total | 32 | 24 | 8 | 220 ± 34.1 | |

[a] Four buffy coat-derived platelet concentrates (PC) from donors with either O RhD positive (O+) or O RhD negative (O-) blood groups comprised the starting material.

MIP-1α, MIP-1β, PDGF-AA, PDGF-AB/BB, TGF-α, TNFα, TNFβ, and VEGF. The concentrations in hPL-PIPL were compared to those in hPL-PIPC and expressed as percentage difference in relation to hPL-PIPC (Eq 1, where $[GF/C]_{hPL-PIPx}$ refers to the concentration of a particular growth factor or cytokine in the hPL-PIPC or hPL-PIPL treatment).

$$\% \ Difference \ = \ \frac{[GF/C]_{hPL-PIPC} - [GF/C]_{hPL-PIPL}}{[GF/C]_{hPL-PIPC}} x \ 100\% \tag{1}$$

## Cell culturing

Mesenchymal stromal cells originating from the bone marrow of two healthy human donors were purchased from Lonza (Walkersville, MD, USA) and stored at -180˚C in liquid nitrogen prior to experimentation. The cells tested negative for viral infections and mycoplasma. Prior to experimentation, the MSCs were cultured in a cell culture medium supplemented with hPL-PIPC through passage 3. Subsequently, the MSCs were distributed into two cell culture flasks supplemented with either hPL-PIPC or hPL-PIPL at the time of cell split (between passage 3 and passage 4). MSCs in passage 5 were used for experimentation.

The culture medium used was a complete cell culture medium (referred to as "expansion medium" in this study) consisting of Dulbecco's Modified Eagle Medium (DMEM) / F12 + Glutamax supplement (Gibco, Grand Island, NY, USA) with 1% penicillin/streptomycin (Gibco) and 2 IU/mL heparin (Leo Pharma A/S, Ballerup, Denmark), supplemented with a sufficient amount of either hPL-PIPC (to produce hPL-PIPC-MSCs) or hPL-PIPL (to produce hPL-PIPL-MSCs) to achieve a final concentration of 9%. Specifically, after allowing solutions to reach ambient temperature, 50 mL of platelet lysate was centrifuged at 4975 × g for 10 minutes and added to 500 mL DMEM / F12 + Glutamax along with 5 mL of penicillin/streptomycin and 200 μL heparin. The medium was allowed to sit for 10 minutes prior to sterile filtration. Sterile filtration was performed using a 0.45 μm low protein-binding funnel (Corning Incorporated, NY, USA) in a closed system. Finally, the medium was aliquoted into 45-mL centrifugation tubes (Corning Incorporated) and stored at -20˚C until use. Once thawed for use in cell cultures, the medium was maintained at 4˚C in a laboratory refrigerator (Angelantoni Life Sciences, Massa Martana, Italy) for a maximum of seven days.

Incubation was done in a Steri-Cult CO2 Incubator, HEPA Class 100 (Thermo Scientific) under the following conditions: 37˚C; 5% $CO_2$, and 95% humidity.

Cell expansion was performed in different vessels appropriately selected for each experiment at a seeding density of 6000 cells/$cm^2$. For the initial cell expansion prior to experimentation, MSCs were expanded in 20 mL of expansion medium in Nunc™ EasYFlask™ 75 $cm^2$ (T75) cell culture flasks (Thermo Fischer Scientific Nunc A/S, Roskilde, Denmark). For long-term proliferation studies, MSCs were expanded in 5 mL of expansion medium in Nunc™ EasYFlask™ 25 $cm^2$ (T25) cell culture flasks (Thermo Fischer Scientific Nunc A/S). In both cases, the expansion medium was replaced every two to three days.

Cell passaging was performed upon reaching 80–90% confluency, as determined visually by daily inspection using a Leica DM IRB inverted contrast microscope (Leica Microsystems, Wetzlar, Germany). In brief, the MSCs were gently washed with 1X PBS (Gibco) and dissociated from the surface in 0.25% 1X Trypsin-EDTA (Gibco) for 5 minutes. Preheated expansion medium was added to neutralize the trypsin-EDTA before the cells were centrifuged at 609 × g for 5 minutes. After centrifugation, the supernatant was discarded and the pellet was carefully resuspended in 1 mL preheated medium prior to cell counting. The cells were diluted 5X by mixing 20 μL resuspended cells, 30 μL 1X PBS, and 50 μL 0.4% trypan blue stain (Gibco) in a

1.5-mL micro tube (SARSTEDT AG & Co., Nümbrecht, Germany). The cell solution was loaded onto a hemocytometer (BRAND GMBH + CO KG, Wertheim, Germany), covered by a glass, and counted at 50X magnification. Viable MSCs were identified by the retention of their round morphology and by their lack of trypan blue uptake. Viable MSCs located in the four corner squares were counted twice using the upper and lower chambers and averaged to estimate the number of cells. If the sum of the four corner squares in a single chamber exceeded 200 viable MSCs, the cell solution was diluted further and the cell count repeated. Cell passaging was completed by seeding 6000 cells/$cm^2$ into a new cell culture vessel.

Long-term proliferation was evaluated by expanding and passaging MSCs in T25 cell culture flasks. Initially, MSCs entering passage 5 were seeded into six T25 cell culture flasks and expanded in cell culture media supplemented with either hPL-PIPC or hPL-PIPL. The expansion medium was replaced every two to three days, and cell passaging was carried out upon reaching 80–90% confluency. All cell culture flasks were passaged on the same day and the number of population doublings (PDs) was determined using Eq 2, where N0 and N1 represent the number of cells seeded and cells harvested, respectively.

$$PDs \;=\; \frac{\log_{10}(N0) - \log_{10}(N1)}{\log_{10}(2)} \tag{2}$$

Cumulative population doublings (CPDs) were expressed as the sum of the PDs obtained in each passage. Cell expansion was terminated upon achieving recovery rates of less than 100% of the seeded cell number after a maximum of 14 days in culture. Daily monitoring was performed to assess morphological alterations.

## Tri-lineage differentiation

MSCs in passage 5 were used to evaluate *in vitro* tri-lineage differentiation potential. Osteogenic and adipogenic differentiation were performed simultaneously using the same cell cultures. Chondrogenic differentiation was performed separately due to the large number of required cells.

## Osteogenic differentiation

Osteogenic differentiation was evaluated at various timepoints during 28 days of stimulation in osteogenic medium. The osteogenic medium consisted of 45 mL DMEM / F12 + Glutamax (Gibco) supplemented with 5 mL platelet lysate, 50 μL dexamethasone (Sigma-Aldrich, St. Louis, MO, USA), 50 μL human/murine/rat BMP-2 (Peprotech, Rocky Hill, NJ, USA), 50 μL L-ascorbic acid (Sigma-Aldrich), and 108 mg β-glycerophosphate disodium salt hydrate (Sigma-Aldrich). An increase in alkaline phosphatase activity and positive staining for mineralization were used as markers. In brief, 3500 cells/$cm^2$ were seeded in quadruplicate in 6-well tissue culture plates (Corning Incorporated) and in triplicate in 12-well tissue culture plates (Corning Incorporated) for each timepoint. Control plates were included by seeding 5500 cells/$cm^2$ in expansion media in absence of osteogenic stimulation. Plates were incubated at 37˚C, 5% $CO_2$, and 95% humidity for up to 28 days. The cell culture medium was replaced every two to three days. The left half of the 12-well plates was used to quantify ALP activity, while the right half was used to detect mineralization by staining with Alizarin Red S.

Enzymatic activity of ALP was evaluated after 7, 14, 21, and 28 days of osteogenic differentiation. Briefly, 0.02% Triton-X (Sigma-Aldrich) diluted in 1X PBS was added to all samples, and then the cells were scraped off the surface and transferred to a 1.5-mL micro tube. The cells were vortexed and then centrifuged at 13,200 × g for 15 minutes at 4˚C. After centrifugation, the supernatant was transferred to a new micro tube and mixed with 500 μL of p-

nitrophenyl phosphate (pNPP) solution, prepared using SIGMAFAST™ pNPP and SIGMA-FAST™ Tris Buffer (Sigma-Aldrich). Next, the solution was incubated for 45 minutes at 37˚C protected from light, and the absorbance was measured at 400 nm. Each sample reading was corrected by the average absorbance of three blank replicates. ALP activity (nmol (p-nitrophe-nol)/min) was calculated using Eq 3,

$$activity \ = \ \frac{^{OD}/_{18.8}}{t} x \ 1000 \ , \tag{3}$$

where OD refers to the optical density obtained at 400 nm (-), 18.8 is the extinction coefficient of p-nitrophenol ($\mu mol^{-1}$), t is time (min), and 1000 is used to convert $\mu mol$ to nmol.

Alizarin Red S staining was performed to visualize mineralization during osteogenic differentiation. Cell cultures were collected and fixed in 4% paraformaldehyde after 7, 14, 21, and 28 days of osteogenic differentiation and stored at 4˚C prior to staining. Cells were washed three times with distilled water ($dH_2O$) before adding a 2% Alizarin Red solution containing Alizarin Red S dye (Sigma-Aldrich) diluted in $dH_2O$. The cells were placed on a rotating shaker and stained for 20 minutes at room temperature, followed by four washing steps using $dH_2O$. The dye was allowed to dry for 24 hours by inverting the plates on paper towels. The following day, images were captured using inverted contrast microscope imaging, and alterations in morphology and formation of bone-like nodules were evaluated.

## Adipogenic differentiation

Adipogenic differentiation was evaluated after 7 and 14 days of stimulation in adipogenic medium. Adipogenic medium consisted of 22.5 mL StemPro® Adipocyte Differentiation Basal Medium (Gibco), 2.5 mL StemPro® Adipogenesis Supplement (Gibco), 0.25 mL penicillin/streptomycin, 2.5 mL platelet lysate, and 20 μL heparin.

In brief, 10,000 cells/cm$^2$ were seeded in triplicate Nunc™ 9 cm$^2$ Slideflasks (Thermo Fischer Scientific Nunc A/S). Control slideflasks were included by seeding 5500 cells/cm$^2$ in expansion media in absence of adipogenic stimulation. Sideflasks were incubated at 37˚C, 5% CO$_2$, and 95% humidity for up to 14 days. The MSCs were allowed to reach a confluency of 50–70% in expansion media prior to introduction of the adipogenic medium. The cell culture medium was replaced every two to three days. Cultures in the 9 cm$^2$ slideflasks were washed three times in 1X PBS and fixed in 3 mL of 4% paraformaldehyde and stored at 4 ˚C after 7 or 14 days prior to Oil Red O staining.

Upon termination of the experiment, the slideflasks were collected and sent to the Department of Pathology (Landspitali, Háskólasjúkrahús, Reykjavík, Iceland) where Oil Red O staining was performed according to departmental protocols. Positive Oil Red O-stained lipid droplets were used as markers of adipogenic differentiation.

## Chondrogenic differentiation

Chondrogenic differentiation was evaluated after 14, 28, and 35 days of stimulation in chondrogenic medium. Chondrogenic medium consisted of 47.9 mL DMEM / F12 + Glutamax supplemented with 9% hPL, 1% penicillin/streptomycin, 50 μL L-ascorbic acid, 50 μL dexamethasone, 500 μL sodium pyruvate (Sigma-Aldrich), 500 μL L-proline (Sigma-Aldrich), 500 μL ITS+ (Gibco) and 5 μL of 10 ng/μL TGF-β3. Production of glycosaminoglycans, as well as positive collagen fibers and lacunae formation visualized by chondrocytic pellet staining with Masson's trichrome were used as markers of chondrogenic differentiation.

In brief, 250,000 cells were seeded in ten 1.5-mL micro tubes containing 0.5 mL chondrogenic media for each timepoint. Control micro tubes were included by seeding 250,000 cells in

expansion media in absence of chondrogenic stimulation. Pellets were formed by centrifugation at $152 \times g$ using a Sorvall Instruments RC5C centrifuge (Thermo Fischer Scientific). The caps were punctured with a sterile needle to allow air exchange, and the tubes were incubated at 37°C, 5% $CO_2$, and 95% humidity for up to 35 days. After 18–24 hours, the tubes were gently agitated to detach the pellets from the wall of the micro tubes. To minimize the stress on the pellets, half of the cell culture medium was replaced every second day. At each sampling time-point (14, 28, and 35 days), three pellets were analyzed for GAG content and two pellets were prepared for histological staining.

To prepare pellets for the GAG assay, the three pellets were pooled into a micro tube containing 500 μL papain extraction reagent (Sigma-Aldrich). The samples were then transferred to a Grant-Bio PHMT heating block (Grant Instruments Ltd, Shepreth, Cambridgeshire, UK) and fully digested at 65°C for a maximum of seven hours. After digestion, the samples were centrifuged at $9660 \times g$ and the supernatants were transferred to new micro tubes and stored at -80°C.

To prepare pellets for histological staining, the pellets were collected and washed in 1X PBS prior to fixation in 0.5 mL 4% paraformaldehyde in new micro tubes. The samples were stored at 4°C prior to histological staining.

The Biocolor Blyscan Sulfated Glycosaminoglycan Assay B1000 (Biocolor Ltd, County Antrim, United Kingdom) was used to quantify the concentrations of GAGs. The assay was performed according to the manufacturer's protocol. In brief, standards diluted in papain extraction reagent were prepared containing GAGs in the working range of 0–5 μg/mL. 1 mL of Blyscan dye reagent was added to a new micro tube prepared for each standard and sample. 100 μL of each standard and sample was then added to the new micro tubes and mixed with the Blyscan dye reagent for 30 minutes on a mechanical shaker (Heidolph, Schwabach, Germany). After incubation, the micro tubes were centrifuged for 10 minutes at $9660 \times g$. The supernatant was separated from the pellet by inverting the micro tubes carefully before adding 0.5 mL dissociation reagent. Prior to quantification, the micro tubes were vortexed to release bound dye into the solution. 200 μL of each standard and sample was loaded into a 96-well tissue culture plate in triplicate and measured at 656 nm using a Multiskan® spectrum spectrophotometer (Thermo Scientific, Vantaa, Finland). The average absorbance of the blank replicates was subtracted from each standard and sample. The GAG concentration of each sample was determined using the standard curve. To express GAG concentration per pellet, the values were divided by three to account for the number of pooled pellets.

Upon experimental termination, the micro tubes were collected and sent to the Department of Pathology (Landspitali, Háskólasjúkrahús, Reykjavík, Iceland). Masson's trichrome and hematoxylin and eosin staining were performed according to departmental protocols.

## Statistical analysis

Statistical comparisons were performed using GraphPad Prism Version 7.04 software (GraphPad Software, Inc., San Diego, CA, USA). Paired t-tests were performed to analyze differences in total protein. Ratio paired t-tests were performed to analyze differences in the content of growth factors and cytokines. Differences in proliferation and differentiation were analyzed using a two-way ANOVA followed by multiple comparisons using Tukey's post hoc test. Differences were considered significant at $p < 0.05$.

The sample size (N) for each experiment refers to the number of experimental units derived from biological units using separate cell culture vessels. Nomenclature and principle were adapted from Lazic et al. [43].

## Acknowledgments

We would like to thank Ragna Landrö for processing the platelet concentrates and Sigrún Bærings Kristjánsdóttir for support with staining procedures.

## Author Contributions

**Conceptualization:** Christian Christensen, Sandra Mjoll Jonsdottir-Buch, Olafur Eysteinn Sigurjonsson.

**Data curation:** Christian Christensen, Sandra Mjoll Jonsdottir-Buch, Olafur Eysteinn Sigurjonsson.

**Formal analysis:** Christian Christensen, Sandra Mjoll Jonsdottir-Buch, Olafur Eysteinn Sigurjonsson.

**Funding acquisition:** Sandra Mjoll Jonsdottir-Buch, Olafur Eysteinn Sigurjonsson.

**Investigation:** Christian Christensen, Sandra Mjoll Jonsdottir-Buch, Olafur Eysteinn Sigurjonsson.

**Methodology:** Christian Christensen, Sandra Mjoll Jonsdottir-Buch, Olafur Eysteinn Sigurjonsson.

**Project administration:** Olafur Eysteinn Sigurjonsson.

**Resources:** Olafur Eysteinn Sigurjonsson.

**Software:** Christian Christensen, Olafur Eysteinn Sigurjonsson.

**Supervision:** Sandra Mjoll Jonsdottir-Buch, Olafur Eysteinn Sigurjonsson.

**Validation:** Christian Christensen, Olafur Eysteinn Sigurjonsson.

**Visualization:** Christian Christensen, Olafur Eysteinn Sigurjonsson.

**Writing – original draft:** Christian Christensen, Sandra Mjoll Jonsdottir-Buch, Olafur Eysteinn Sigurjonsson.

**Writing – review & editing:** Christian Christensen, Sandra Mjoll Jonsdottir-Buch, Olafur Eysteinn Sigurjonsson.

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
