## [Decision Letter · Decision Letter 0]

11 Sep 2019

PONE-D-19-18862

Effects of amotosalen treatment on human platelet lysate bioactivity

PLOS ONE

Dear Dr Sigurjonsson,

Thank you for submitting your manuscript to PLOS ONE. After careful consideration, we feel that it has merit but does not fully meet PLOS ONE’s publication criteria as it currently stands. Therefore, we invite you to submit a revised version of the manuscript that comprehensively addresses the points raised by both referees during the review process.

We would appreciate receiving your revised manuscript by Oct 26 2019 11:59PM. To enhance the reproducibility of your results, we recommend that if applicable you deposit your laboratory protocols in protocols.io, where a protocol can be assigned its own identifier (DOI) such that it can be cited independently in the future. For instructions see: http://journals.plos.org/plosone/s/submission-guidelines#loc-laboratory-protocols

We look forward to receiving your revised manuscript.

Kind regards,

Andre van Wijnen

Academic Editor

PLOS ONE

Journal Requirements:

1. Thank you for including your competing interests statement; "The authors have declared that no competing interests exist."

We note that one or more of the authors are employed by a commercial company:

Platome Biotechnology, Hafnarfjörður, Iceland

Reviewers' comments:

Reviewer's Responses to Questions

**Comments to the Author**

1. Is the manuscript technically sound, and do the data support the conclusions?

Reviewer #1: Partly

Reviewer #2: Yes

2. Has the statistical analysis been performed appropriately and rigorously? 

Reviewer #1: Yes

Reviewer #2: Yes

3. Have the authors made all data underlying the findings in their manuscript fully available?

Reviewer #1: Yes

Reviewer #2: Yes

4. Is the manuscript presented in an intelligible fashion and written in standard English?

Reviewer #1: Yes

Reviewer #2: Yes

5. Review Comments to the Author

Reviewer #1: Christensen and colleagues report results of an ex-vivo study of bone-marrow derived MSCs expansion and differentiation with human platelet lysates (hPL) with two different pathogen inactivation processes. Specifically, this study addresses the issue that clinical application of MSCs usually required and in vitro expansion step for clinically relevant numbers, but this is done with supplementation of basal mammalian cell culture medium with growth factors. The concept of avoiding using supplements with animal substances, hPL produced from expired and pathogen inactivated platelet concentrates can be used in place of fetal bovine serum. However globally, most transfusion units are not pathogen inactivation. They aimed to assess the quality of hPL produced from expired platelet concentrates with pathogen inactivation applied after platelet lysis (compared to conventional after collection), as well as its ability to support MSC proliferation and tri-lineage differentiation.

They compared results of expanding and differentiation bone marrow MSCs using hPL derived from pathogen inactivated platelet lystes (with pathogen inactivation after lysis of expired platelets) versus using hPL produced from conventional expired pathogen inactivated human platelet concentrates (with pathogen inactivation applied soon applied soon after blood donation). THEY FOUND THAT PATHOGEN INACTIVATION AFTER LYSIS HAD LOWER CONCENTRATIONS OF SOLUBLE GROWTH FACTOR AND CYTOKINES THAN THOSE WITH CONVENTIONAL PATHOGEN INACTIVATION, AND THAT IN CELL CULTURE THE MSCS PROLIFERATED AT A REDUCED BUT MORE CONSISTENT RATE THAN THE CONVENTIONAL INACTVATION. THEY FOUND THE ABILITY TO SUPPORT TRILINEAGE DIFFERENTIATION WAS COMPARABLE BETWEEN THE LYSATES.

They interpreted their results as supportive of hPL being produced from expired and untreated platelet lysates by post lysis pathogen inactivation for global hPL production methods to increase the pool of starting material and meet future demand for animal-free supplements in human cell culturing.

The writing is clean, the methods address the aims, the results clear, the table and figures well done and interpretable, but the interpretation of the results is overreaching and the lack of comparison to the standard of fetal bovine serum disappointing.

Major issues:

1. The very large ranges in the % differences in the growth factors between the two batches of PIPC and PIPL makes it hard to know what a representative ‘baths’ of these hPL really are. I believe to really make any comment on the concentrations of growth factors in this setting, and to really interpret the results, more than 2 batches must be compared. Not only are there differences between the two hPL pathogen inactivation techniques, but also between the batches of the same technique…. This is a problem for interpreting the results.

2. It is disappointing that there is not a control with the standard fetal bovine serum. Given the literature suggests that conventional pathogen inactivated hPL is as good or better than standard fetal bovine serum, it would be important to compare this method to the conventional AND post-lysate pathogen inactivation methods. This is especially important given the concentrations of selected soluble growth factors were lower in the post-lysate pathogen inactivation group, because it is important to know if these concentrations are of equivalence to the standard bovine fetal serum (as there may be a ceiling effect in the amount of growth factor needed for proliferation and differentiation, further supporting less importance on the growth factor differences and more importance on the biologic effects)

3. Along these lines, the study is missing a limitations section that should be added, and should include discussion of the important issues with changing from a non-human source to a human blood-product source need to be mentioned as the pathogen reduction techniques may not completely mitigate the risk of viral transmissions.

4. The interpretation of the results should be tempered. Given there were lower concentrations of key growth factors, lower cumulative population doublings, no comparison to standard fetal bovine serum, and no study of retention of genomic stability of avoidance of tumorigenicity, it is unclear to me whether this successfully supports long term cell proliferation.

5. Please discuss the decision to store for 3 weeks of -80 degrees. Why was this chosen? The LOS at -80 may effect results and needs to be studied going forward.

Minor issues:

1. Please spell out the growth factors in the manuscript and in the table, they cannot be assumed to be known.

Reviewer #2: Overall the study titled “effects of amotosalen treatment on human platelet lysate bioactivity” by Christensen et al. is sound and reveals interesting results. However, we believe the authors should address the following concerns:

1. All the experiments were done using fresh platelets, but authors let the platelets expire. We think that experiments using the banked expired platelets should be performed to complement current findings.

2. In every section of the results, author should provide the rationale for each of the experiments, what they did, explain the results and their inferences, as much as possible and a brief manner.

3. In equations 1 and 3, authors should “write” multiplication symbol instead of using a “dot”.

3. In every place authors should correct CO2/CO2 (should be CO2).

6. PLOS authors have the option to publish the peer review history of their article (what does this mean?). If published, this will include your full peer review and any attached files.

Reviewer #1: No

Reviewer #2: No

---

## [Author Response · Author response to Decision Letter 0]

29 Nov 2019

Below are responses to the reviewer’s comments

Reviewer 1:

Comment 1: The very large ranges in the % differences in the growth factors between the two batches of PIPC and PIPL makes it hard to know what a representative ‘baths’ of these hPL really are. I believe to really make any comment on the concentrations of growth factors in this setting, and to really interpret the results, more than 2 batches must be compared. Not only are there differences between the two hPL pathogen inactivation techniques, but also between the batches of the same technique…. This is a problem for interpreting the results.

Response: We thank the reviewer for pointing this out. We agree with this comment and a line has been included to clarify that the individual difference of the two batches and the implication on the ability to generalize is a limitation of the study (page 12, line 213). It would indeed have been interesting to assess the concentrations of growth factors using a larger pool of batches, however for this proof-of-concept study we were only able to produce platelet concentrates from a total of 32 buffy coats. We have taken this into account for future studies.

Comment 2: It is disappointing that there is not a control with the standard fetal bovine serum. Given the literature suggests that conventional pathogen inactivated hPL is as good or better than standard fetal bovine serum, it would be important to compare this method to the conventional AND post-lysate pathogen inactivation methods. This is especially important given the concentrations of selected soluble growth factors were lower in the post-lysate pathogen inactivation group, because it is important to know if these concentrations are of equivalence to the standard bovine fetal serum (as there may be a ceiling effect in the amount of growth factor needed for proliferation and differentiation, further supporting less importance on the growth factor differences and more importance on the biologic effects).

Response: We thank the reviewer for addressing this. This is a valid point and we fully agree that a comparison to fetal bovine serum is valuable. However, for this proof-of-concept study where the central focus was to assess whether it was possible to produce platelet lysates by performing pathogen inactivation post-expiry, we believe that including fetal bovine serum would be slightly out of scope. The direct comparison between hPL and fetal bovine serum has been carried out in several papers including two by our own research group. See:

https://www.ncbi.nlm.nih.gov/pmc/articles/PMC5506010/

https://www.ncbi.nlm.nih.gov/pubmed/25198449

https://www.ncbi.nlm.nih.gov/pubmed/23874839

Comment 3: Along these lines, the study is missing a limitations section that should be added, and should include discussion of the important issues with changing from a non-human source to a human blood-product source need to be mentioned as the pathogen reduction techniques may not completely mitigate the risk of viral transmissions.

Response: We agree and a line in the introduction has been added accordingly, clarifying that the INTERCEPT™ Blood System does not guarantee 100% pathogen reduction and is indeed bypassed by non-enveloped viruses such as hepatitis A and hepatitis E (page 4, line 54).

Comment 4: The interpretation of the results should be tempered. Given there were lower concentrations of key growth factors, lower cumulative population doublings, no comparison to standard fetal bovine serum, and no study of retention of genomic stability of avoidance of tumorigenicity, it is unclear to me whether this successfully supports long term cell proliferation.

Response: We agree with the reviewer that the results in the submitted state did not fully support the conclusion that hPL-PIPL supports long-term cell proliferation alone. We refrained from the wording “successful” and added a line in the discussion pointing out to fully elucidate the effect on long-term cell proliferation, assessment of genomic stability and tumorigenicity is important (page 13, line 225). We also added to the conclusion to clarify that hPL-PIPL is solely viewed as a results of a possible production method and may be a valuable tool moving forward, but the results do not support the notion that hPL-PIPL is better or worse than conventional hPL-PIPC. Further studies are needed to follow up on this.

Comment 5: Please discuss the decision to store for 3 weeks of -80 degrees. Why was this chosen? The LOS at -80 may effect results and needs to be studied going forward.

Response: This is a very important point. We do not have reason to believe that this had a negative impact on the quality of the end product as it has been discussed as general practice by several groups (https://www.ncbi.nlm.nih.gov/pubmed/29071726). For this study, we decided upon these storage conditions due to practical reasons. By the time the platelet concentrates were ready to be processed by the National Blood Bank, we were awaiting the arrival of the first donor cells. Optimally, we would have preferred the avoid storage at all, but it was not possible.

Comment 6: Please spell out the growth factors in the manuscript and in the table, they cannot be assumed to be known.

Response: We thank the reviewer for this suggestion. We have spelled out the growth factors in the manuscript.

\f

Reviewer 2:

Comment 1: All the experiments were done using fresh platelets, but authors let the platelets expire. We think that experiments using the banked expired platelets should be performed to complement current findings.

Response: We thank the reviewer for this comment. It is important for us to stress that the platelet concentrates used in the study were all banked and allowed to expire prior to experimentation. We have previously compared fresh platelets to expired platelets (https://www.ncbi.nlm.nih.gov/pubmed/23874839) but this was out of the scope of this study. Rather we wanted to study the possibility of producing viable platelet lysates from expired platelet concentrates applying pathogen inactivation post-expiry. We hope this answers the reviewer’s concern.

Comment 2: In every section of the results, author should provide the rationale for each of the experiments, what they did, explain the results and their inferences, as much as possible and a brief manner.

Response: We agree with this comment. However, we believe that the rationales of the experiments are well-described in the discussion section of the paper. E.g. at page 11, line 199 we argue that growth factors within the α-granules are important for MSC proliferation and differentiation and provided the basis for quantifying soluble growth factors and cytokines. At page 12, line 216 we discuss the impact of “higher” concentrations of growth factors on MSC proliferation supporting the use of the cumulative population doubling assay. Similar rationale is provided for each of the differentiation assays throughout the discussion.

Comment 3: In equations 1 and 3, authors should “write” multiplication symbol instead of using a “dot”.

Response: We thank the reviewer for this suggestion. We have replaced the use of a “dot” with the multiplication symbol “x” in equation 1 and 3 at page 16, line 311 and page 20, line 401. 

Comment 4: In every place authors should correct CO2/CO2 (should be CO2).

Response: We thank the reviewer for this observation. It is correct that we falsely used the term “CO2” instead of the correct form “CO2”. This has been corrected at all places.

---

## [Decision Letter · Decision Letter 1]

6 Feb 2020

PONE-D-19-18862R1

Effects of amotosalen treatment on human platelet lysate bioactivity

PLOS ONE

Dear Dr Sigurjonsson,

Thank you for submitting your revised manuscript to PLOS ONE. Your paper was re-reviewed by the original reviewers and one of them feels you have adequately addressed their comments, but Reviewer#1 has residual concerns and recommends against acceptance of your work. Because this decision is split, we are providing you with a second opportunity to revise this study with the guidance of a second set of comments from the reviewers. 

We would appreciate receiving your revised manuscript by Mar 22 2020 11:59PM. To enhance the reproducibility of your results, we recommend that if applicable you deposit your laboratory protocols in protocols.io, where a protocol can be assigned its own identifier (DOI) such that it can be cited independently in the future. For instructions see: http://journals.plos.org/plosone/s/submission-guidelines#loc-laboratory-protocols

We look forward to receiving your revised manuscript.

Kind regards,

Andre van Wijnen

Academic Editor

PLOS ONE

Reviewers' comments:

Reviewer's Responses to Questions

**Comments to the Author**

1. If the authors have adequately addressed your comments raised in a previous round of review and you feel that this manuscript is now acceptable for publication, you may indicate that here to bypass the “Comments to the Author” section, enter your conflict of interest statement in the “Confidential to Editor” section, and submit your "Accept" recommendation.

Reviewer #1: (No Response)

Reviewer #2: All comments have been addressed

2. Is the manuscript technically sound, and do the data support the conclusions?

Reviewer #1: No

Reviewer #2: Yes

3. Has the statistical analysis been performed appropriately and rigorously? 

Reviewer #1: Yes

Reviewer #2: Yes

4. Have the authors made all data underlying the findings in their manuscript fully available?

Reviewer #1: Yes

Reviewer #2: (No Response)

5. Is the manuscript presented in an intelligible fashion and written in standard English?

Reviewer #1: Yes

Reviewer #2: Yes

6. Review Comments to the Author

Reviewer #1: Christensen and colleagues have submitted a revision of their reported results of an ex-vivo study of bone-marrow derived MSCs expansion and differentiation with human platelet lysates (hPL) with two different pathogen inactivation processes. Specifically, this study addresses the issue that clinical application of MSCs usually require an in vitro expansion step for clinically relevant numbers, but this is done with supplementation of basal mammalian cell culture medium with growth factors. Addressing the concept of avoiding using supplements with animal substances, hPL produced from expired and pathogen inactivated platelet concentrates can be used in place of fetal bovine serum. However, globally most transfusion units are not pathogen inactivated, so they aimed to assess the quality of hPL produced from expired platelet concentrates with pathogen inactivation applied AFTER platelet lysis (compared to conventional after collection), as well as its ability to support MSC proliferation and tri-lineage differentiation.

They compared results of expanding and differentiation of bone marrow MSCs using hPL derived from pathogen inactivated platelet lystes (with pathogen inactivation after lysis of expired platelets) versus using hPL derived from conventional expired pathogen inactivated human platelet concentrates (with pathogen inactivation applied soon applied soon after blood donation). THEY FOUND THAT PATHOGEN INACTIVATION AFTER LYSIS HAD LOWER CONCENTRATIONS OF SOLUBLE GROWTH FACTOR AND CYTOKINES THAN THOSE WITH CONVENTIONAL PATHOGEN INACTIVATION, AND THAT IN CELL CULTURE THE MSCS PROLIFERATED AT A REDUCED BUT MORE CONSISTENT RATE THAN THE CONVENTIONAL INACTVATION. THEY FOUND THE ABILITY TO SUPPORT TRILINEAGE DIFFERENTIATION WAS COMPARABLE BETWEEN THE LYSATES. They interpreted their results as supportive of hPL being produced from expired and untreated platelet lysates (by post lysis pathogen inactivation) for global hPL production methods to increase the pool of starting material and meet future demand for animal-free supplements in human cell culturing.

Unfortunately, based on my prior review, very few changes have been made in the manuscript (only a few short sentences added and a couple of words changed), and their responses do not adequately address my concerns. They have now called this study a ‘proof-of-concept’ study in the reviewer responses, (but this is mentioned only once in their discussion). I would suggest, that if this is indeed a ‘proof-of-concept’ study, this be clarified throughout the ENTIRE manuscript including adding those words to the title so this is not misleading to readers. As previously mentioned, the findings are not interpretable without more than two batches due to the very large ranges in the % differences in the growth factors between the two batches of PIPC and PIPL (not only are there differences between the two hPL pathogen inactivation techniques, but also between the batches of the same technique), and the missing bovine fetal serum comparison group (given the citations of direct comparisons performed between hPL and bovine fetal serum by this author group, there should be no reason why this cannot be performed for this study). The couple short sentences added to the manuscript without a focus on a full limitations section (as previously suggested), will not suffice given the significant limitations in current form. Given the many limitations, these should be addressed together in a true limitations section and furthermore need to include the issue raised of storage for 3 weeks at -80 especially given it was decided based on practical reasons.

Reviewer #2: (No Response)

7. PLOS authors have the option to publish the peer review history of their article (what does this mean?). If published, this will include your full peer review and any attached files.

Reviewer #1: No

Reviewer #2: No

---

## [Author Response · Author response to Decision Letter 1]

24 Feb 2020

Below are responses to the reviewer’s comments on the revised manuscript

Reviewer 1:

Comment 1: Unfortunately, based on my prior review, very few changes have been made in the manuscript (only a few short sentences added and a couple of words changed), and their responses do not adequately address my concerns

Response: Based on this and Reviewer 1’s further comments on the revised manuscript, we have further expanded upon the concerns raised. Further details on specific issues are described below.

Comment 2: They have now called this study a ‘proof-of-concept’ study in the reviewer responses, (but this is mentioned only once in their discussion). I would suggest, that if this is indeed a ‘proof-of-concept’ study, this be clarified throughout the ENTIRE manuscript including adding those words to the title so this is not misleading to readers

Response: Thank you for pointing out that we needed to clarify that this is a “proof-of-concept” study throughout. As suggested, we have reworked the title and it now specifically references that this is a proof-of-concept work (page 1). We have also made sure to make this clear in the abstract (page 2, line 27), in the introduction (page 4, line 95), in multiple places in the discussion section (page 12, line 205; page 12, line 210; page 13 line 232; page 14, line 271), and in the conclusion (page 15, line 285). 

Comment 3: . As previously mentioned, the findings are not interpretable without more than two batches due to the very large ranges in the % differences in the growth factors between the two batches of PIPC and PIPL (not only are there differences between the two hPL pathogen inactivation techniques, but also between the batches of the same technique), and the missing bovine fetal serum comparison group (given the citations of direct comparisons performed between hPL and bovine fetal serum by this author group, there should be no reason why this cannot be performed for this study)..

Response: This large differences between the two batches does indeed make it difficult to generalize the results of this study to hPL in general. We believe that, as this was a proof-of-concept study (unfortunately we did not make this clear in the original submission), that our results provide enough evidence to suggest that the application of pathogen inactivation after platelet lysis is worth further study. This has now been more explicitly stated in the manuscript (page 14, lines 274-282), and we have also clarified the limitations resulting from the small number of replicates (pages 12-13, lines 232-234; page 14, lines 271-274)

With regard to a comparison with FBS, we have added to the discussion a note that it was not included in this work because previous studies have shown hPL to be comparable to FBS as a cell culture supplement (page 12, lines 209-211). We have also noted that future work should include an FBS control (page 14, lines 279-280). Unfortunately, none of the studies references specifically looked at growth factor and cytokine levels in the FBS supplements used, so we cannot provide a comparison within our manuscript to previous results. We don’t believe that it is warranted to analyze the levels of these factors in FBS at this point, considering the study is already complete, but, again, we have noted that this should be included in future work.

Comment 4: ). The couple short sentences added to the manuscript without a focus on a full limitations section (as previously suggested), will not suffice given the significant limitations in current form. Given the many limitations, these should be addressed together in a true limitations section.

Response: In order to address this concern, in addition to the sentences previously added into the manuscript after the initial review, we have added a paragraph at the end of the discussion section which discusses how the limitations inherent in this proof-of-concept work should be addressed in future work. Please note that the main limitations are explicitly stated within this discussion. (page 14, lines 274-282)

Comment 5: [The limitations] need to include the issue raised of storage for 3 weeks at -80 especially given it was decided based on practical reasons. (comment base on the initial submission: Please discuss the decision to store for 3 weeks of -80 degrees. Why was this chosen? The LOS at -80 may effect results and needs to be studied going forward.)

Response: Again, thank you for raising this important point. As we initially stated, we do not have reason to believe that this had a negative impact on the quality of the end product as it has been discussed as general practice by several groups (https://www.ncbi.nlm.nih.gov/pubmed/29071726). For this study, we decided upon these storage conditions due to practical reasons: by the time the platelet concentrates were ready to be processed by the National Blood Bank, we were awaiting the arrival of the first donor cells. Optimally, we would have preferred to avoid storage at all, but it was not possible.

We would also like to point out that the production of platelet lysates does involve temperature cycling from -80°C to 37°C three times, as stated in the Materials and methods section (lines 321-322). Therefore, it is not unreasonable to store the cells at -80°C, given that this temperature is used for lysis. In addition, it must be noted that even in the absence of any storage period, the cells would have been exposed to -80°C during the process of lysis, and therefore we do not have any reason to believe that the relatively brief storage period used would have had any detrimental effect on the platelets.

Finally, it is worth noting that both treatments were exposed to the -80°C storage conditions.

---

## [Decision Letter · Decision Letter 2]

25 Mar 2020

Effects of amotosalen treatment on human platelet lysate bioactivity: a proof-of-concept study

PONE-D-19-18862R2

Dear Dr. Sigurjonsson,

We are pleased to inform you that your manuscript has been judged scientifically suitable for publication and will be formally accepted for publication once it complies with all outstanding technical requirements.

With kind regards,

Andre van Wijnen

Academic Editor

PLOS ONE

Additional Editor Comments (optional):

Reviewers' comments:

Reviewer's Responses to Questions

**Comments to the Author**

1. If the authors have adequately addressed your comments raised in a previous round of review and you feel that this manuscript is now acceptable for publication, you may indicate that here to bypass the “Comments to the Author” section, enter your conflict of interest statement in the “Confidential to Editor” section, and submit your "Accept" recommendation.

Reviewer #1: All comments have been addressed

2. Is the manuscript technically sound, and do the data support the conclusions?

Reviewer #1: Yes

3. Has the statistical analysis been performed appropriately and rigorously? 

Reviewer #1: Yes

4. Have the authors made all data underlying the findings in their manuscript fully available?

Reviewer #1: Yes

5. Is the manuscript presented in an intelligible fashion and written in standard English?

Reviewer #1: Yes

6. Review Comments to the Author

Reviewer #1: Christensen and colleagues have submitted a revision of their reported results of an ex-vivo study of bone-marrow derived MSCs expansion and differentiation with human platelet lysates (hPL) with two different pathogen inactivation processes.

On their last revision, very few of my comments had been addressed. I appreciate that they have addressed that and have reframed the study throughout as a proof of concept study. By reframing, the issue of not being able to generalize these results beyond the two batches, and the missing bovine fetal serum comparison group are much improved. The expanded limitations section is much appreciated.

7. PLOS authors have the option to publish the peer review history of their article (what does this mean?). If published, this will include your full peer review and any attached files.

Reviewer #1: No

---

## [Editor Report · Acceptance letter]

1 Apr 2020

PONE-D-19-18862R2 

Effects of amotosalen treatment on human platelet lysate bioactivity: a proof-of-concept study 

Dear Dr. Sigurjonsson:

I am pleased to inform you that your manuscript has been deemed suitable for publication in PLOS ONE. Congratulations! Your manuscript is now with our production department. 

With kind regards,

on behalf of

Dr. Andre van Wijnen 

Academic Editor

PLOS ONE